# Frog Crabs (*Ranina ranina*) in South Penghu Marine National Park, Taiwan: A Case Study of Population Dynamics and Recreational Fishing Sustainable Development

Chun-Han Shih 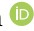

Department of Leisure & Tourism Management, Shu-Te University, Kaohsiung 82445, Taiwan; f92b45028@ntu.edu.tw; Tel.: +886-76158000 (ext. 3412)

**Abstract:** The frog crab/red frog crab (*Ranina ranina*), a species of symbolic significance in the South Penghu Marine National Park, Taiwan, represents a collaboration between marine conservation and recreational fishing under Sustainable Development Goal 14 (SDG14) as defined by the United Nations. From 2020 to 2021, the growth and reproduction of *R. ranina* were examined in the Taiwan Strait, off the coast of Taiwan. Samples were gathered from the South Penghu Marine National Park water square in Penghu County using red frog crab nets. A comparative analysis of the existing biological literature has revealed that the spawning season of *R. ranina* differs among populations, as evidenced by varying percentages of ovigerous females: 10–90% in Hachijojima, Japan; 86% in Molokai, Hawaii; 1–17% in the Andaman Sea, Thailand; more than 50% in Mindanao, Philippines; and 30–80% in New South Wales, Australia, and Taiwan. Additionally, analysis of the reproductive patterns, growth parameters, and spawning seasons of *R. ranina* can serve as a scientific foundation for the implementation of SDG14 as well as the formulation of conservation principles for resource management. This research has underscored the essential role of localized conservation strategies that cohesively resonate with broader global sustainability goals, offering a strategic framework for effective marine resource management.

**Keywords:** marine protected area; *Ranina ranina*; sustainable development goal; recreational fishing

## 1. Introduction

The Decapoda Brachyura *Ranina ranina* (Linnaeus, 1758), commonly known as the red frog crab, ranks among the largest crabs and is a favored commercial species in areas like Southeast Asia, Japan, China, Taiwan, the Philippines, and Australia. Though not naturally widespread in Taiwan, it commands a relatively high market price [1,2]. Red frog crab fisheries are spread throughout the Indo-Pacific, with Australia responsible for the most catches [3–6]. Worldwide, many regions have established aquaculture fisheries for this crab due to its appearance and color. In Chinese tradition, it is considered a symbol of blessings and used in offerings. The crab exhibits a backward walking pattern alluded to by the term "regressive snoring" and is characterized by a vermilion shell, short and hairy horns, flat legs, and seven abdominal segments that resemble those of Garfield crabs. With a reddish body, it inhabits offshore mud and sand. Recent overfishing has led to dwindling resources for frog crabs, resulting in smaller specimens (with a maximum carapace length of 15 cm and a maximum male weight of 0.9 kg) in Taiwan compared to other regions. Fortunately, the Marine National Park Headquarters (MNPH) in southern Penghu Lake, Taiwan's ninth national park and second marine-based park, was established on 18 October 2014, after preparation since 2010. Activities involving investigation, research, and the monitoring of natural and human resources have been initiated there, including the formation of a long-term ecological research network to enhance marine ecological diversity conservation. This also involves the restoration of critical ecosystems and populations that are endangered or have been destroyed. Penghu Island is one of the significant benchmarks

for recreational fisheries in Asia, attracting millions of tourists annually, with an overall production value of 200 million USD [7]. In cooperation with the South Penghu Marine National Park, the Penghu County Government has established the strategy of sustainable fishing under SDG14 for environmental justice. This collaboration has established the goal of achieving a balance between industrial development and environmental protection in Penghu, defending the sovereignty of the fishing industry, and realizing recreational fishing under the sustainability of the ocean. The recognition of *R. ranina* as a flagship species in the South Penghu Marine National Park (SPMNP) has offered an opportunity to align marine conservation with SDG14. This paper investigates the development of sustainable marine resource policies to support this alignment, addressing both ecological conservation and socioeconomic sustainability. Consequently, the MNPH plays direct and indirect roles in protecting *R. ranina*. Local fishing communities and the government are increasingly focusing on fishery restoration, with many Marine Protected Areas (MPAs) allowing limited biota extraction despite only offering partial protection [7–9]. Additionally, it is noteworthy that the sample area of this study represents unexplored research concerning global fishery ecological data that are related to *R. ranina*.

Currently, length frequency analysis stands as one of the most reliable and accessible methods for estimating the growth and mortality parameters of crabs. It can be broken down into two main classifications; the first includes statistical decomposition techniques like MIX [10] and MULTIFAN [11], and the second entails mode progression approaches such as Electronic Length Frequency Analysis (ELEFAN) [12]. While the former offers a more intensive statistical perspective, the latter tends to be more arbitrary. The von Bertalanffy Growth Equation (VBGE) has remained a viable approximation for Penaeidae, even though crustacean growth is sporadic during molting [13–15]. Essential to the assessment and management of crustacean stock, dynamic pool models of yield per recruit (Y/R) include certain biological parameters; for instance, those of growth [16]. The challenge of estimating growth arises from crabs losing their exoskeletons when molting, which has rendered conventional methods ineffective in determining their aging. The only precise means to gauge crustacean growth and mortality parameters is length frequency analysis [17–19]. As first-grade crabs typically undergo asynchronous molting, the use of the VBGE to derive an average length for this age group is uncomplicated [15,20–22]. Furthermore, water temperature exerts a notable influence on crustacean growth [2,15–23]. Pauly and Gaschutz (1979) [24] modified the basic VBGE model to incorporate seasonal fluctuations, enhancing the detailed comprehension of crab growth. This amended VBGE model is predominantly used in studies of crustacean stock [21,22].

In Taiwan, the western coastline features significant crab fisheries. Species like *Portunus trituberculatus* [25] and *Portunus pelagicus* [26] have been the subjects of growth and fishery biology research. Despite the abundance and commercial significance of *R. ranina*, its fishery biology has remained unexplored [23]. The study of ELEFAN in exploited fisheries can enhance the comprehension of its dependability, potentially boosting confidence in its application in data-limited fisheries. The study in this document employed ELEFAN techniques to identify the recruitment pattern and growth parameters of *R. ranina* around the Marine National Park Headquarters on Taiwan's western shoreline, as accurate appraisals of the growth, mortality, and additional population parameters of safeguarded species within national park marine reserves are indispensable for correct evaluation and stewardship.

## 2. Materials and Methods

### 2.1. Study Area

The research area for this study is situated in the southern region of Penghu, encompassing Dongjiyu, Xijiyu, Dongyu Pingyu, and Xiyu Pingyu and commonly referred to as the Penghu South Four Islands National Park Reserve. Positioned roughly between 119°30′ and 119°41′ east longitude and between 23°14′ and 23°16′ north latitude, this zone includes not only the aforementioned four main islands but also neighboring islands and reefs, such

as Turban, Anvil, Zhongzi, Zhumajiao, and Hutouyu, as shown in Figure 1. Making up the Southern Four Islands of Penghu are the Dongji Islet, the Xiji Islet, the Dongyuping Islet, and the Xiyuping Islet. This region is part of the SPMNP, Taiwan's ninth national park, and is well-known for its rich coral reef ecosystem, making it a premier location for snorkeling and diving within Penghu. Based on the field investigation of water temperature in this study, the average sea temperature in the waters of the SPMNP may vary with seasonal changes. During the summer, the average water temperature may range between 28 and 30 degrees Celsius, while in winter, it may range from 14 to 24 degrees Celsius.

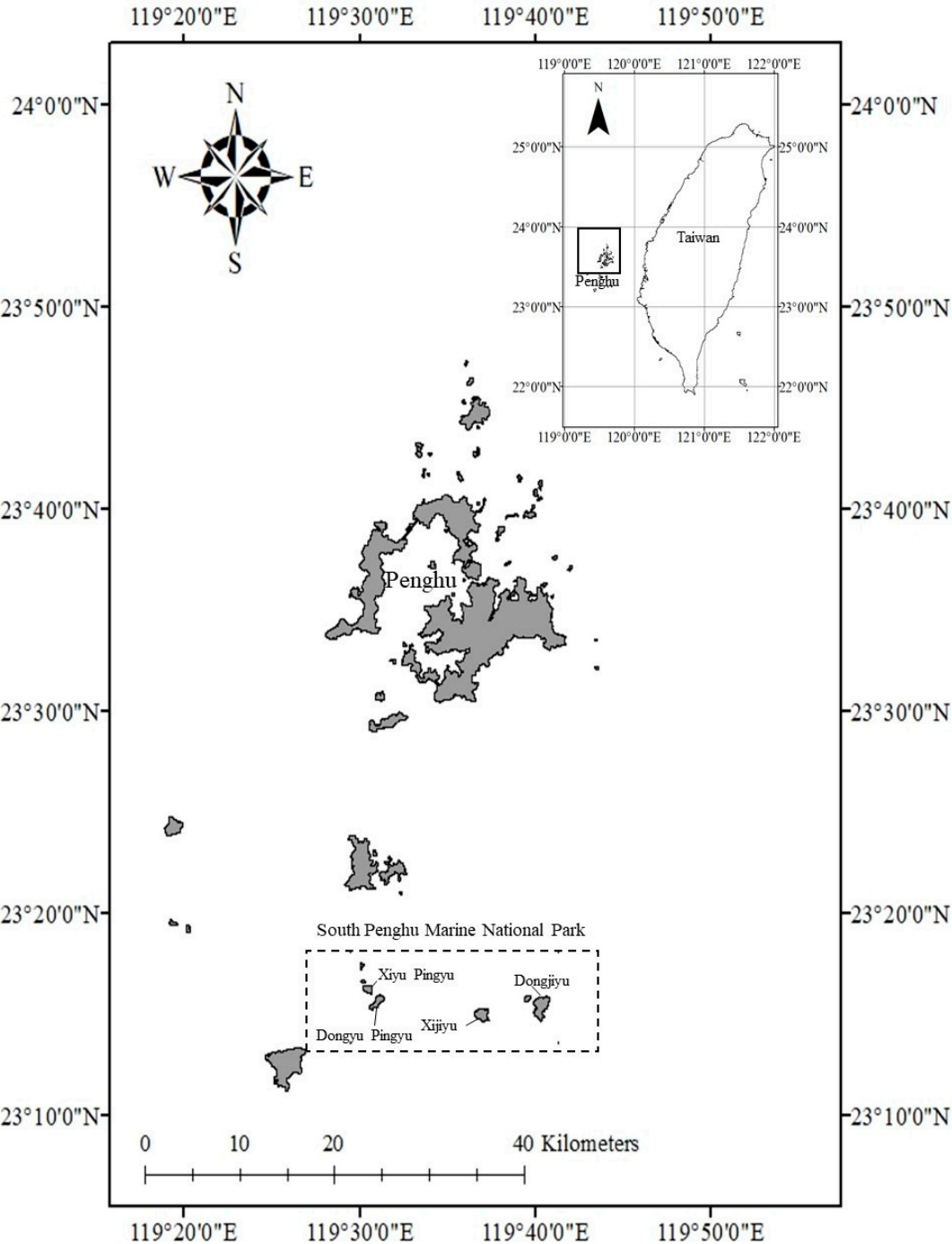

**Figure 1.** The shadowed area shows the sampling area in the South Penghu Marine National Park water square.

## 2.2. Biological Sampling

Between September 2020 and August 2021, length frequency data were collected every month from specific beam trawler monitoring samples on the western coast of Taiwan (as shown in Figure 1). The mesh size of the codend of the beam trawler was 18 mm. For each sample, the sex was identified, the Carapace Length (CL) was accurately measured to the closest 0.1 of a mm, and the Body Weight (WT) was determined to the nearest 0.01 of a g. The collected samples included 742 females and 473 males. The measurement of each carapace extended from the middle of the back edge to the front tip of the rostrum (i.e., the length of the medulla oblongata's carapace). The carapace length data, segregated by sex and based on 1 mm intervals, were analyzed using ELEFAN within the FiSAT software [27].

## 2.3. Biological Characteristics

The equation $WT = a\,CL^b$ outlines the relationship between CL and WT; "b" denotes the growth exponent and "a" is a constant factor. We carried out regression analysis on the log-transformed values of WT and CL, followed by computation of the back-transformed power functions for both females and males in relation to WT versus CL. To represent the growth of the crabs, a seasonally fluctuating model of the VBGE was utilized. Initially developed in [24], it was subsequently revised in [28]. The seasonal VBGE is (1)

$$L_t = L_\infty(1 - exp(-k(t - t_0) + (CK/2\pi)sin2\pi(t - t_s) + (CK/2\pi)sin2\pi(t_0 - t_s))) \quad (1)$$

where Lt signifies the length at time t, $L_\infty$ represents the asymptotic length, and K is the growth coefficient. C denotes the amplitude of seasonal changes, varying between 0 and 1, and $t_s$ refers to the fraction of the year in which the growth rate peaks. The ELEFAN I program also identified a parameter referred to as the Winter Point (WP), which symbolizes the time fraction during the period of slowest growth, expressed as $WP = t_s + 0.5$. The parameter $t_0$ indicates the theoretical age at a length of zero. In this study, the K values for the female and male crabs were calculated to be 0.29 and 0.23 (1/year), respectively, based on [29].

## 2.4. Reproduction Analysis

The chi-square ($X^2$) test was used to verify the sex ratio at a 95% confidence level. The quantitative Gonadosomatic Index (GSI) was computed with the following equation: GSI = (gonad weight in grams/body weight in grams) × 1000. This is as referenced in [30]. The males were categorized into mature and immature classes. The spawning pattern modes, or reproductive cycles, were assessed by estimating the average monthly index values over specified time frames (of months). Quantitative and qualitative evaluations of the gonadosomatic index supplemented the analysis of the gonadal stages. The proportion of sexually mature females in each size class was calculated based on the counts of egg-laying females and those in the second phase of ovarian development. For the sexually mature females (P), a logarithmic curve was constructed corresponding to the CL ratio using the following Formula (2) from [31]:

$$P = 1/(1 + exp(-(a + b \times CL))) \quad (2)$$

where a and b are defined parameters. Correlation analysis of the linearized variables P and CL was used to perform parameter estimation, with the carapace length of a female that is 50% sexually mature ($CL_{50}$) determined as (a/b). A range of techniques facilitated the analysis of the length frequency data to deduce growth parameters and mortality rates.

## 3. Results and Discussion

### 3.1. Biological Survey and Population Structure

The CLs and WTs ranged from 53.39 mm to 151.25 mm (mean ± SD, 77.58 ± 13.91 mm) and 67.7 g to 1401.5 g, respectively, in the males (3), while in the females (4), these values ranged from 58.5 mm to 126.9 mm (67.91 ± 16.26 mm) and 59.1 g to 744.1 g, respectively,

as shown in Figure 2. Following are the linear regression equations that best describe the relationship between carapace width (CW) and CL:

$$\text{Male, } CL = 10.2474 + 1.0112\, CW \ (r = 0.97, N = 473), \tag{3}$$

$$\text{Female, } CL = -2.2322 + 0.8858\, CW \ (r = 0.94, N = 742). \tag{4}$$

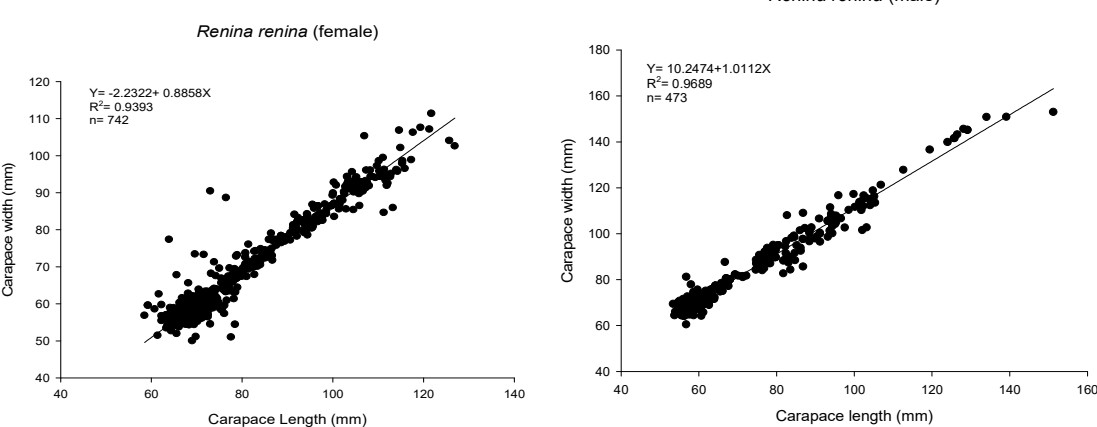

**Figure 2.** Relationship between CL and CW for females (**left**) and males (**right**).

As shown (5) and (6) in the power equation, CL and WT have the following relationship, also depicted in Figure 3:

$$\text{Male, } BW = 0.008\, CL^{2.8895} \ (r = 0.98, N = 473), \tag{5}$$

$$\text{Female, } BW = 0.0002\, CL^{3.1332} \ (r = 0.93, N = 742). \tag{6}$$

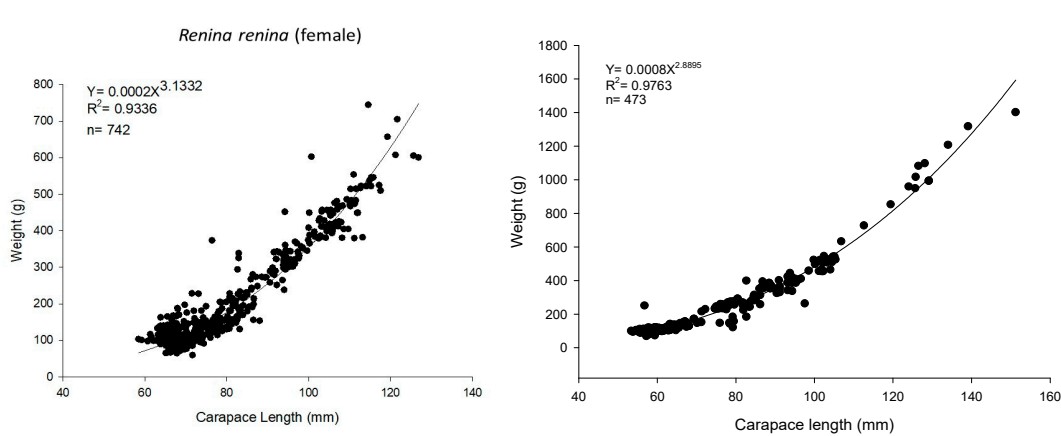

**Figure 3.** Relationship between CL and WT for females (**left**) and males (**right**).

According to an ANCOVA, there was a significant difference between the slopes of the regression lines for the males and the females ($p < 0.05$).

These initial $L_\infty$ values were separately seeded into ELEFAN I to produce the optimized seasonal growth curve. In Figure 4, the length frequency and seasonal growth curves of both the females (7) and the males (8), restructured using ELEFAN I, are displayed. The males and females exhibited the following seasonal growth parameters:

$$\text{Female: } L_\infty = 152.00 \ (\text{mm}), K = 0.38, C = 0.95, WP = 0.05, \tag{7}$$

$$\text{Male: } L_\infty = 151.25 \text{ (mm)}, K = 0.32, C = 0.95, WP = 0.05. \tag{8}$$

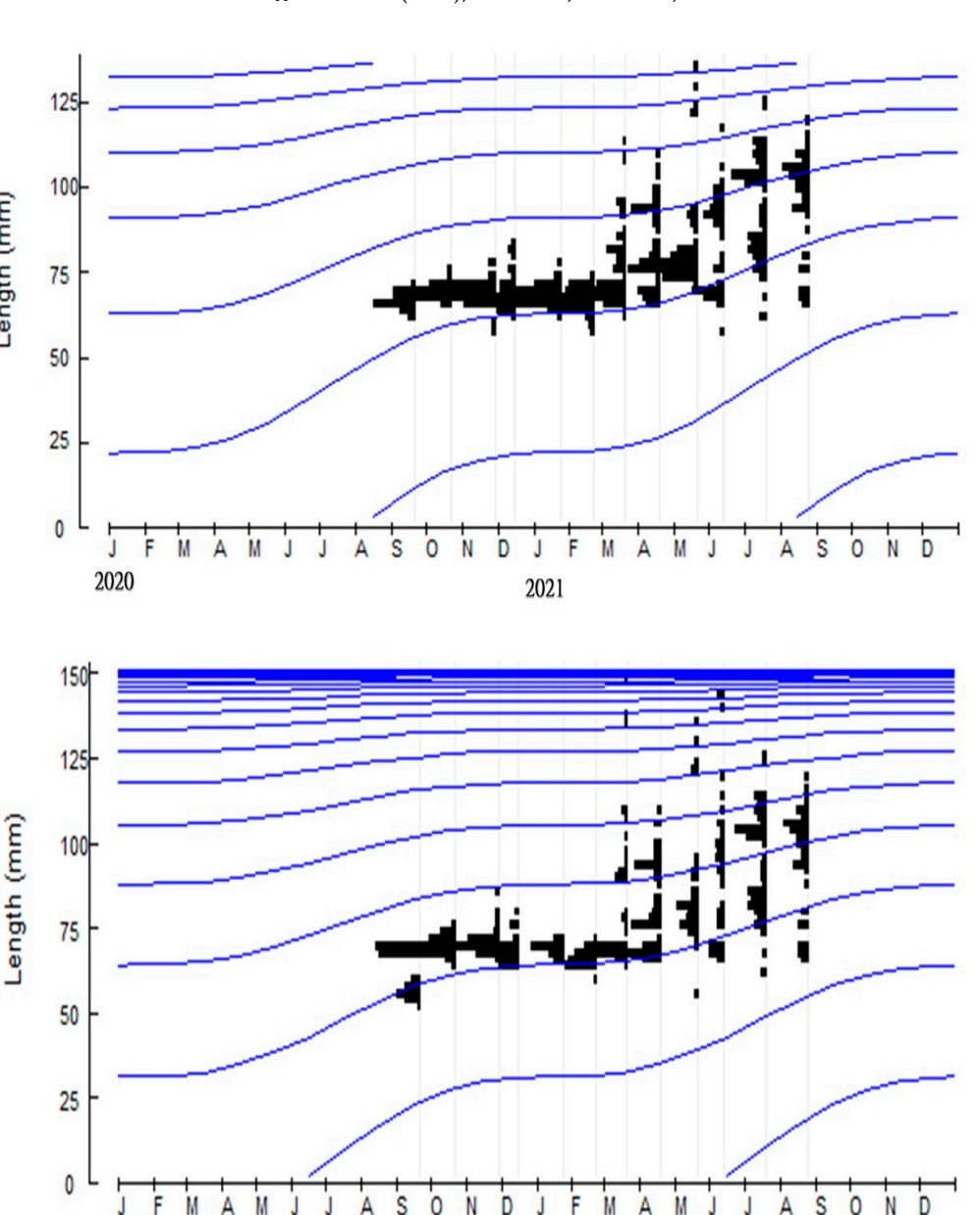

**Figure 4.** Restructured length frequency data and seasonal growth curves for females (**top**) and males (**bottom**). Numbers indicate sample size.

As the recruiter patterns were analyzed with ELEFAN II, one or two normal distributions were automatically matched. In this calculation, as the value was zero, determining the exact time of recruitment was not possible. Figure 5 shows recruitment peaks of approximately equal strength for *R. ranina*. Size at maturity is a very important life history trait for this and other crustaceans. Based on the data shown in Figure 6, the estimated size of a female at 50% sexual maturity ($CL_{50}$) was 7.02 cm $\pm$ 0.56 cm $CL_{12}$. This means that half of females reach sexual maturity when their carapace lengths are around 7 cm. This information can help to determine the minimum legal size for harvesting *R. ranina* and protect the reproductive potential of the population.

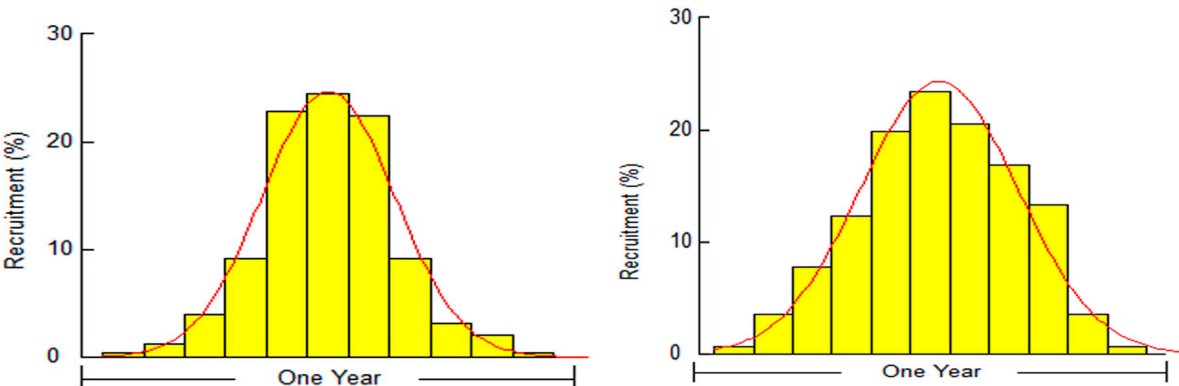

**Figure 5.** Recruitment patterns for females (**left**) and males (**right**).

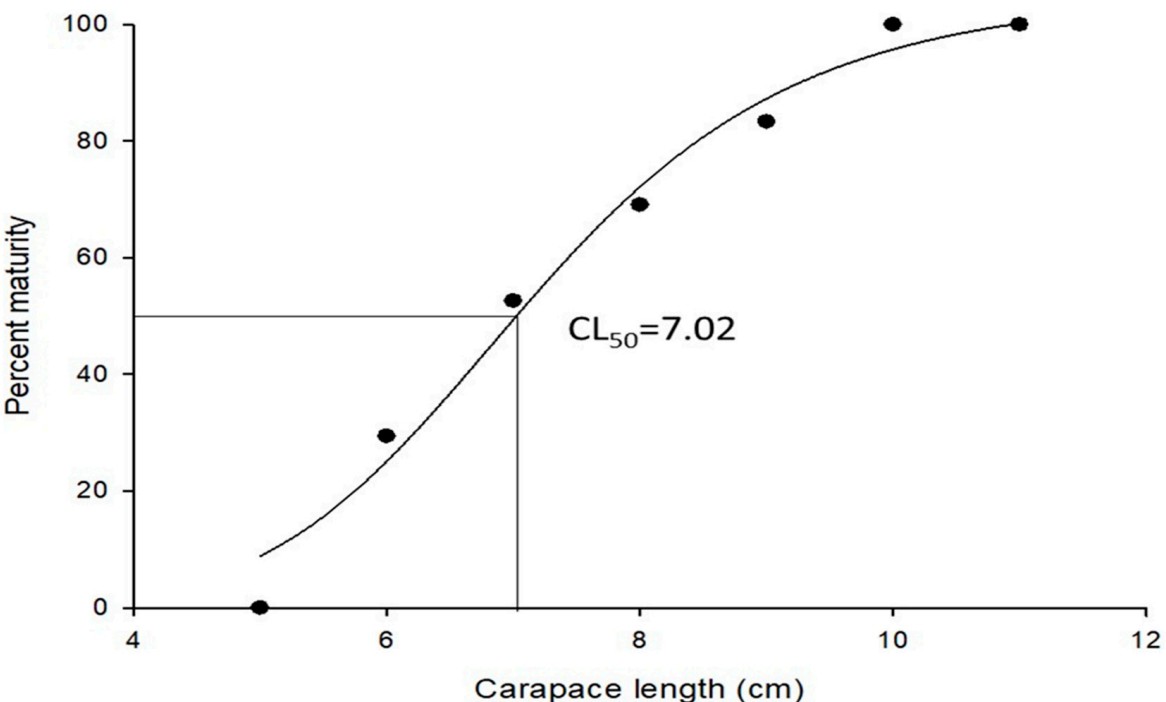

**Figure 6.** A logarithmic function fitting the proportion of mature female *Ranina ranina* to CL (7.02 cm), which corresponds to a proportion of 0.5 (50% of females are mature).

### 3.2. Reproduction Biology

In the female crabs, the ovaries predominantly exhibited stages III and V, with the average GSI fluctuating between 1.38 and 1.93 from June to the end of November. From November 2020 to May 2021, the majority of the ovaries advanced to stage IV or V. The GSI peaked at 1.93 in November 2020 and decreased to 1.57 in May 2021. It also experienced a minor decline between December and April, which coincided with ova release. The testes reached full maturity between January and March, with an average GSI ranging from 0.45 to 0.97, as shown in Figure 7b. The 473 female *R. ranina*, ranging between 5.77 and 11.73 in CL, were analyzed using the following logarithmic function to determine the correlation between the CL and the proportion of sexually mature females within the 1 mm CL group:

$$P = 1/(1 + exp\,(104.85 - 5.9525CL))$$

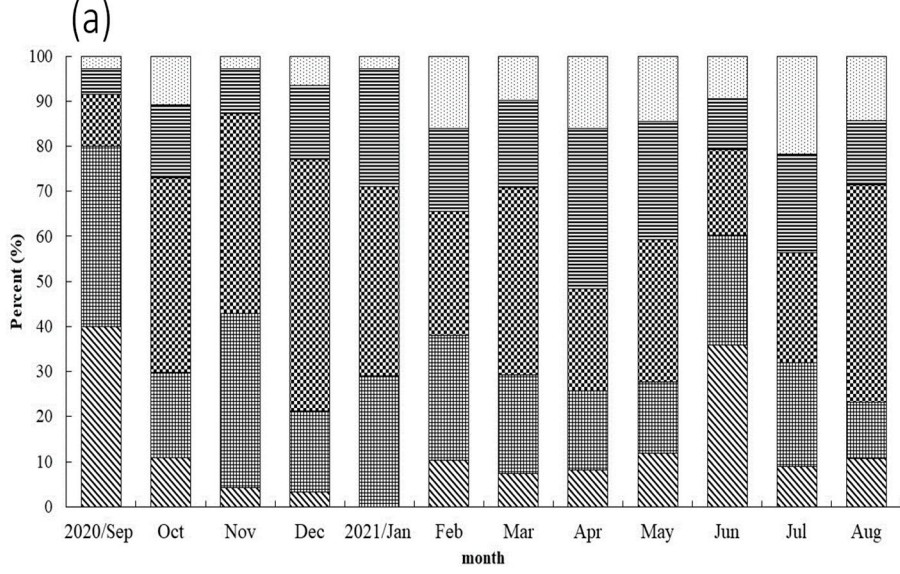

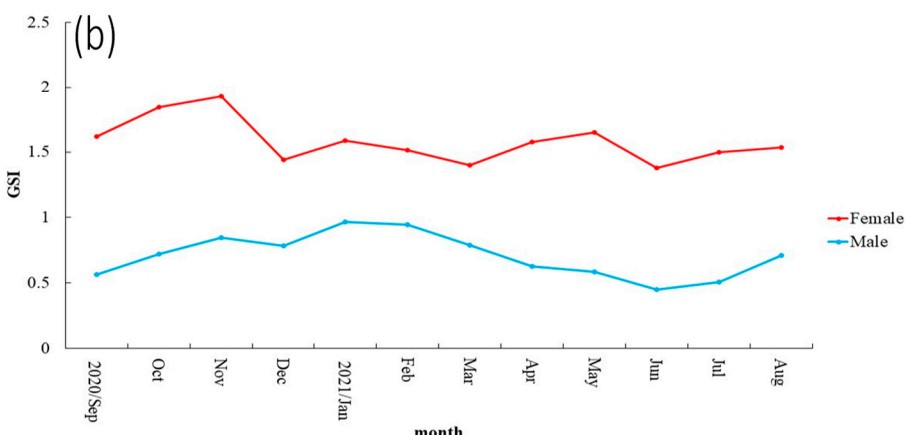

**Figure 7.** (**a**) Percentages of female maturity stages and (**b**) and mean GSIs of females and males of *R. ranina* from September 2020 to August 2021.

Thus, the estimated size of a female at 50% sexual maturity (±95% confidence interval) was 7.02 cm ± 0.56 cm CL, as shown in Figure 6.

This computation estimated the size of a female at 50% sexual maturity to be a 7.02 cm ± 0.56 mm CL. The yearly fluctuating male-to-female sex ratio deviated widely from 1:1, reaching as much as 1:1.06. This overall discrepancy was statistically significant, with females being 1.5 times more prevalent. However, males outnumbered females almost twofold in October 2020 (Figure 8). This study operated under three foundational assumptions, (i) an assortment of lengths was encapsulated within the samples; (ii) the input was a significant factor in the selection of the gears; and (iii) over a time span greater than one month, the total sample size would result in approximately equal monthly sample sizes. Hoenig et al. (1987) [32] introduced a methodology for assessing the adequacy of length frequency data that are pertinent to the analysis of both growth and mortality. Such an analysis is deemed satisfactory or excellent if the sample size accumulated over six months exceeds 500 or 1000, respectively. Various factors have supported this approach, including (i) the total length variation between the sexes, spanning from 60.7 to 133.1 mm for females and 52 to 94 mm for males and encapsulating the entire range of lengths [33]; (ii) the inherent restrictions of mesh size selection within crab studies, such as when one investigation employed only an 18 mm mesh size [14]; and (iii) cumulative sample sizes of 1058 for females and 639 for males over six consecutive months. The collected length frequency

datasets in this study are congruent with the aforementioned assumptions, rendering them suitable for the estimation of growth parameters and mortality rates. Wolf (1989) [34] asserted that in length frequency data, certain modal groups should be discernible, with modal lengths that vary temporally. Those characteristics were also observed in this study (Figure 5). While this study met several standards for the application of length frequency data in the population parameter estimation, the utilization of only twelve months of such data may have led to the omission of essential length distribution details, diminishing the reliability of the estimate. However, the presence of larger individuals in species with lifespans exceeding one year can partially compensate for this information loss. Thus, this error may not be of grave concern. Future research would benefit from more exhaustive and extended length frequency data for enhanced reliability in population parameter estimation. The potential for minor errors in the application of adult growth parameters to length frequency data or the selection of length frequency distribution affecting the recruitment pattern estimation notwithstanding [21,35], the ELEFAN method has demonstrated notable flexibility and robustness in growth and mortality analysis. Thorough comprehension of the biology of the species in question is vital for the effective application of this method, though it can be employed provisionally if such knowledge is lacking [12].

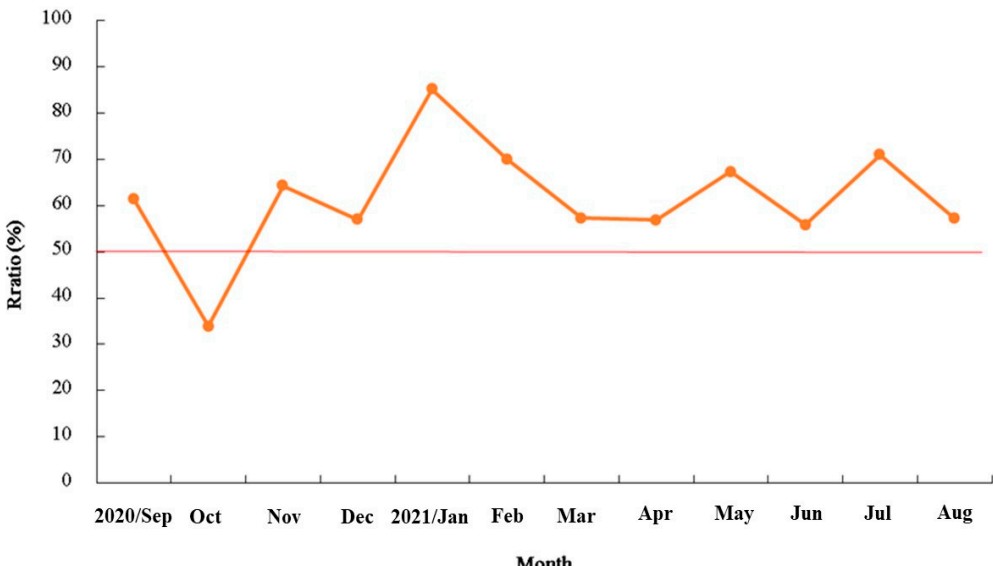

**Figure 8.** Sex ratio of males and females of *R. ranina* from 2020 to 2021.

The integrity of length and frequency approximations can be enhanced by concurrently examining several length frequency datasets [11]. Commonly employed methods for analyzing multiple length frequency datasets include Shepherd's length composition analysis [23], ELEFAN [12], MULTIFAN [11], and the projection matrix method [24]. Only ELEFAN or MULTIFAN can calculate seasonal growth parameters in species that demonstrate such patterns. While MULTIFAN typically produces more accurate results [25] and necessitates less input and fewer presuppositions than ELEFAN [12], it is often employed to study the length and frequency of a single recruitment. However, this method may not be applicable to crabs, as they typically exhibit two recruitment pulses per year, as seen in species like *Callinectes sapidus* [26], *Portunus trituberculatus* [25], and *Portunus pelagicus* [15]. As a result, the updated MULTIFAN software advises doubling the time interval between two length samples to acknowledge biannual recruitment pulses, subsequently doubling the final K estimate to ascertain the yearly value [27]. In spite of the apparent modes of the two given recruitment pulses, the time between them may not be six months. MULTIFAN cannot, therefore, be used to estimate the growth parameters of some crab species without solid ground for assuming recruitment pulses every six months. This assumption is not necessary for ELEFAN, and its estimates are comparable with those from MULTIFAN [27].

Even when there are only one or two recruitment pulses per year, the estimates drawn from ELEFAN have been reliable [12]. Analysis of length data requires consideration of the applicability of the length frequency data used, which has been evaluated using a number of criteria.

In this study, we observed that the reproductive maturity period of *R. ranina* in Taiwan was from January to June, which was quite different from those of the crabs in Japan [36], the Philippines, Thailand, and Australia [37]. The spawning seasons of the different *R. ranina* populations vary considerably, according to surveys that have indicated distinct proportions of ovigerous females per month. Studies from New South Wales, Australia, and Taiwan have shown 30–80% ovigerous females monthly, whereas Japan has noted 10–90%, Hawaii has observed 86%, Thailand has observed 1–17%, and the Philippines have documented over 50% [38,39]. *R. ranina*'s breeding season differs across regions (Figure 9). In Japan, 10–90% of females were ovigerous from May to September, peaking in June. Similarly, 86% of females were ovigerous from May to September in Hawaii, with none outside those months. In Thailand, 1.1–16.6% of females carried eggs from November to May, peaking from November to February [30]. In Australia, 30–80% of females were ovigerous in December, although bimonthly sampling has suggested ovigerous females from November to January [40,41]. Generally, spawning occurs during warmer months, from October to February [42]. Peak ovigerous seasons are similar across tropical regions but differ between the temperate regions of Australia, Japan, and Hawaii, indicating that seasonality depends on ocean conditions beyond temperature, including salinity, light, current, and larval food availability [43].

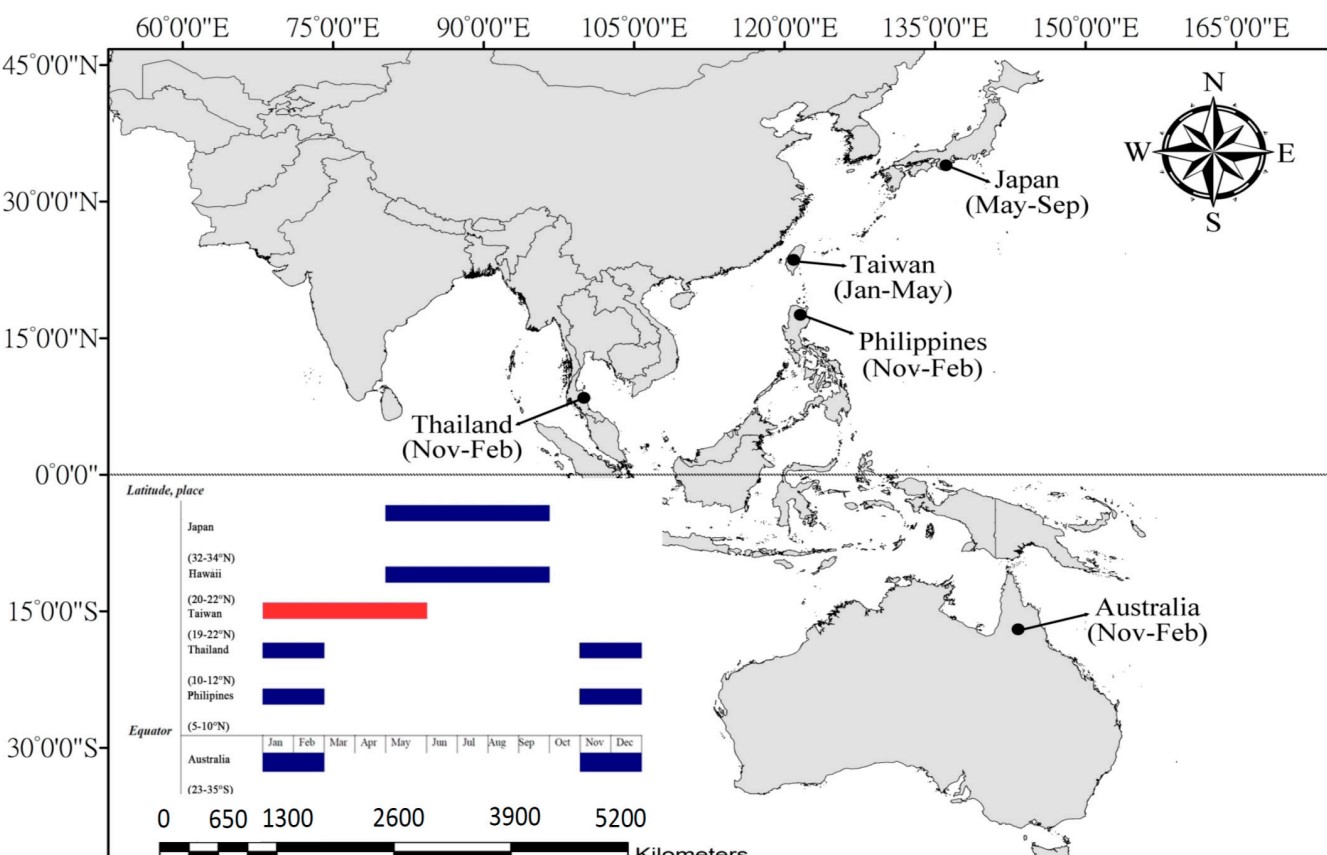

**Figure 9.** Variations in spawning seasons of *R. ranina* across different populations, highlighted by percentages of ovigerous females: 10–90% in Hachijojima, Japan; 86% in Molokai, Hawaii; 1–17% in the Andaman Sea, Thailand; over 50% in Mindanao, Philippines; and 30–80% in New South Wales, Australia, and Taiwan.

*3.3. Frog Crabs and the Impact of Recreational Fishing: A Policy Relationship*

The sustainable management of marine resources is crucial for the health of global oceans and the prosperity of reliant communities. The frog crab, as a flagship species in the SPMNP, stands out as a critical focus of study. Flagship species are charismatic representatives of ecosystems that engage the public, thus driving conservation efforts. The uniqueness and ecological importance of the frog crab have magnified its significance within the SPMNP. Its role is pivotal in heightening public awareness about marine ecosystem vulnerabilities and value. Moreover, its place in the food chain bolsters marine stability. Studying both farmed crab development and marked wild crabs is crucial for devising and validating techniques to determine crab ages. A significant relationship is observed between variations in growth parameters and population characteristics like instantaneous natural mortality. Thus, a reliable estimation of natural mortality is essential, with an urgent need to further explore other population dynamics [44,45]. For instance, there is a pressing need for specific yield models and length-based model evaluations to compute crab recruitment and abundance. Considering the climate challenges, the SPMNP is confronted with the relentless rise of extreme weather, as island regions are increasingly susceptible to these abrupt shifts. These include rising sea levels, heightened storm activity, ecological changes, seawater intrusion, and the pronounced impacts of recreational fishing tourism during summer, leading to phenomena like coral bleaching.

In addressing these challenges, the SPMNP partnered with the Penghu County Government. Together, in 2020, they initiated a drive to boost environmental education policies with ocean sustainability at its core. Responding to the global vision of net-zero emissions by 2050, efforts were made to uphold the ecological balance of Penghu City. An emphasis on "sustainable island management" was prioritized. Penghu City's alignment with international and national sustainability goals was assessed, focusing on sustainable consumption, production, and climate action strategies. To diminish bottled water consumption, Penghu City introduced the "Offer Tea Action APP," leading to significant carbon footprint reduction. The fisheries sector introduced incentives for fishermen to cease fishing during *R. ranina*'s breeding season, particularly in the SPMNP's core conservation areas. This led to a significant reduction in fishing activities. Comprehensive policies were enacted to embrace sustainable marine practices, incorporating aspects like resource recycling, clean water, and emphasis on controlled recreational fishing tourism to protect marine habitats. Owing to collaborative efforts, numerous environmental education initiatives have been launched, focusing primarily on recreational fishing and its impacts. These initiatives have reached a broad audience, with digital platforms like Facebook and Line enhancing outreach. The policy approach for *R. ranina* within the SPMNP can be summarized as follows:

(i)    Monitoring and Research: detailed analysis is vital to understand the frog crab's population dynamics, habitat needs, and potential threats, ensuring that recreational fishing adopts a sustainable approach.

(ii)   Community Engagement: engaging local communities in conservation activities will make them proactive partners in safeguarding the frog crab and broader marine ecosystems.

(iii)  Integrative Approach: adopting a holistic strategy that includes other species ensures that the focus on frog crabs do not overshadow other biodiversity concerns.

(iv)   Global Collaboration: partnering with international organizations and neighboring countries to share best practices and resources ensures a unified effort in marine conservation.

(v)    Policy Review and Update: regularly revisiting and updating policies related to the frog crab, in light of new scientific findings and societal needs, guarantees the strategies remain effective.

In conclusion, the frog crab's role in the SPMNP provides a compelling model for sustainable marine management, highlighting the species' potential to garner public support while emphasizing the importance of robust scientific methods for lasting conservation success.

## 4. Conclusions

Our examination of *R. ranina* within the SPMNP in Taiwan underscores the importance of understanding the delicate balance between recreational fishing, local conservation strategies, and global sustainability goals. The frog crab, both as an emblem of ecological health and for its economic significance in East Asia, symbolizes the intricate challenges of marine biodiversity preservation. Essential findings highlight the continuous need for scientific research focusing on the species' biological dynamics, growth patterns, and reproductive characteristics. Collaboration with governmental entities has led to targeted initiatives such as elective fishing restraint during breeding seasons, reduction of plastic waste, and enduring island management practices to protect marine habitats for both conservation and recreational purposes. Monitoring trends in farmed crabs and studying tagged wild crabs are pivotal for developing and validating age determination techniques. Distinct connections between perceived variances in growth traits and population metrics, like instantaneous natural mortality, underscore the necessity for a reliable natural mortality assessment. There' is an immediate call for deeper investigations into diverse population dynamics, including yield models tailored to individual recruitment and length-based model analyses to estimate crab recruitment and population density. Recreational fishing data indicates decreased average sizes of both male and female crabs in 2021 compared to the preceding year. The extended spawning season from November to February witnesses a surge in maturing male crabs from September to December. Morphological insights, such as carapace length at initial maturity and monthly gender ratio shifts offer a more granulated comprehension. It is also noteworthy that the spawning season for *R. ranina* varies regionally. In summation, the protection and management of frog crab populations in the SPMNP serve as an exemplar for the sustainable stewardship of marine resources, harmonizing conservation needs with recreational fishing interests.

**Funding:** This work was supported by the National Science and Technology Council: NSTC 107-2221-E-236-002. The funders had no role in the study design, data collection and analysis, decision to publish, or preparation of this manuscript.

**Institutional Review Board Statement:** Not applicable.

**Informed Consent Statement:** Not applicable.

**Data Availability Statement:** Not applicable.

**Acknowledgments:** We thank the Global Institute for Green Tourism at the University of California, Berkeley, for supporting the cruises of the biological survey. We would also like to thank our anonymous reviewers, whose useful suggestions were incorporated into this manuscript.

**Conflicts of Interest:** The authors declare no conflict of interest.

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
