# Peer review of "Frog Crabs (Ranina ranina) in South Penghu Marine National Park, Taiwan: A Case Study of Population Dynamics and Recreational Fishing Sustainable Development"

_water, doi:10.3390/w15203689_

Round 1
Reviewer 1 Report (Previous Reviewer 2)
The authors significantly improved the quality of the manuscript and addressed most of the issues. I think the manuscript could be accepted and considered for publication.
This manuscript is greatly improved over the original version. The authors have satisfactorily addressed all my comments and suggestions in the first review. Thank you!
From my side, I would say it is ready to go online just after following a few minor corrections.
Some specific suggestions are:
1. Increase the quality of the figure
2. Figure 3 (left): Replace “Renina renina” to “Renina renina”
3. Reference: Some references are not followed the journal style. For example, no. 1, 2, 8, 10, 12, 16, 21, 24, and 27.
Please carefully format all the references according to the journal style and if possible provide DOI number.
4. Another thorough English check would be beneficial.
Thank you for inviting me to review the manuscript.
Best wishes
I think the quality of English is acceptable.
Thank you
Author Response
Ms. Ref. No.: water-2649568
Title: Frog Crabs (Ranina ranina) in South Penghu Marine National Park, Taiwan: A Case Study of Population Dynamics and Recreational Fishing Sustainable Development
Reviewer #1: Comments to water-2649568 by Shih C.H:
Comment on MS:
1.Increase the quality of the figure.
Ans: Thank you for your comment. I have corrected.
2.Figure 3 (left): Replace “Renina renina” to “Renina renina”
Ans: Thank you for your comment. I have corrected.
3.Reference: Some references are not followed the journal style. For example, no. 1, 2, 8, 10, 12, 16, 21, 24, and 27. Please carefully format all the references according to the journal style and if possible provide DOI number.
Ans: Thank you for your comment. I have corrected.
4.Another thorough English check would be beneficial.
Ans: Thank you for your comment. I have already viewed English.
All [page number] and [line number] represented below are referred to the revised manuscript.
Reviewer 2 Report (New Reviewer)
In my opinion, the manuscript "Frog Crab (Ranina ranina) in South Penghu Marine National Park, Taiwan: Aligning Conservation with Sustainable Development Goal 14 for Marine Resource Policy" is well done, well written and, excluding a a modest revision, can be published in Water.
Below are some suggestions:
Line 28 – “The frog crab (Ranina ranina), commonly known as the red frog crab (Linnaeus, 1758), ..”, change in: The Decapoda Brachyura Ranina ranina (Linnaeus, 1758), commonly known as the red frog crab (Linnaeus, 1758), ..
Figure 5 - I recommend carrying the same scale on the abscissae for males and females (25 or 30 for both graphs)
Figure 7, if the relevant result is shown before Figure 6, then 7 should be named 6 and inserted before
Line 210 - mm?
Figure 6 - insert "a" and "b" in the two parts of the figure and make the caption included in "a" more readable.
Author Response
Ms. Ref. No.: water-2649568
Title: Frog Crabs (Ranina ranina) in South Penghu Marine National Park, Taiwan: A Case Study of Population Dynamics and Recreational Fishing Sustainable Development
Reviewer #2: Comments to water-2649568 by Shih C.H:
Comment on MS:
- Line 28 – “The frog crab (Ranina ranina), commonly known as the red frog crab (Linnaeus, 1758), ..”, change in: The Decapoda Brachyura Ranina ranina(Linnaeus, 1758), commonly known as the red frog crab (Linnaeus, 1758), .. .
Ans: Thank you for your comment. I have corrected.
- Figure 5 - I recommend carrying the same scale on the abscissae for males and females (25 or 30 for both graphs)
Ans: Thank you for your comment. I have corrected.
- Figure 7, if the relevant result is shown before Figure 6, then 7 should be named 6 and inserted before
Ans: Thank you for your comment. I have corrected.
- Line 210 - mm?.
Ans: Thank you for your comment. I have corrected.
- Figure 6 - insert "a" and "b" in the two parts of the figure and make the caption included in "a" more readable.
Ans: Thank you for your comment. I have corrected.
All [page number] and [line number] represented below are referred to the revised manuscript.
This manuscript is a resubmission of an earlier submission. The following is a list of the peer review reports and author responses from that submission.
Round 1
Reviewer 1 Report
Lines 64-65: At present, length frequency analysis is possibly the sole method that is both accessible and trustworthy for calculating the growth and mortality parameters of crab.
This is not true. Tagging experiments are employed to determine growth and mortality. E.g., Diele K., V. Koch, 2010. Growth and mortality of the exploited mangrove crab Ucides cordatus (Ucididae) in N-Brazil. Journal of Experimental Marine Biology and Ecology, 395: 171-180. https://doi.org/10.1016/j.jembe.2010.08.029.
Kristin Windsland, Total and natural mortality of red king crab (Paralithodes camtschaticus) in Norwegian waters: catch–curve analysis and indirect estimation methodsICES Journal of Marine Science, Volume 72, Issue 2, January/February 2015, Pages 642–650. https://doi.org/10.1093/icesjms/fsu138
Line 193: What are the initial Lꝏ values mentioned here?
Figure 7: The carapace length corresponding to 50% maturity is symbolized as CW50. In the caption, CW50 is not mentioned. There is no consistency in using CW and CL in the text.
Lines 179-180 (Equations (3) and (4), you have determined the relationships between CW and CL. However, I cannot see where you have used CW in the analysis. What is the purpose of determining these relationships?
Line 203: It is not recruitment model, but recruitment pattern as implemented in FiSAT software.
Line 256: It is not correct to state that the value is zero. What you have done was that as t-zero was knot known, it was not possible to determine the exact time of recruitment.
The caption of Figure 5: In the figure, two panels are not given as 'top' and [down' panels.
There is no connectivity of estimation of growth, recruitment patterns, size of maturity, etc. You mentioned estimated values of mortality (Lines 259, 309-310). However, I cannot see that you have estimated mortality.
Section 3.3 is entirely alien in the manuscript. The information given in this section is nt supported by the analysis, and as such, it remains anecdotal.
Section 5 (Conclusion) is not supported by the results of the analysis.
There are many places where extensive language improvements are needed.
The paper is not suitable for publication in its present form.
The marked MS is attached herewith.

There are many places where extensive language improvements are needed.
Author Response
Dear Reviewer,
I sincerely thank the Editors and Reviewers for their detailed feedback and suggestions, which have significantly improved my paper. I'm especially grateful to Reviewer #1, Reviewer 2, Reviewer 3, and Reviewer 4 for their constructive and encouraging comments. I appreciate the careful consideration of my manuscript and the valuable feedback provided.
I have thoroughly revised the paper based on all the comments and suggestions. I would now like to resubmit the research article titled “Frog Crabs (Ranina ranina) in South Penghu Marine National Park, Taiwan: Aligning Conservation with Sustainable Development Goal 14 of Marine Resource Policy” by Shih, C-H., for consideration in WATER. I'm deeply appreciative of the feedback from the anonymous referees, which has been instrumental in refining my manuscript. I've addressed each comment, and the changes are detailed in the following sections.
Sincerely yours,
Chun-Han Shih
Author to whom correspondence should be addressed;
E-Mail: f92b45028@ntu.edu.tw ;
Tel: +886–7-6158000 ext 3412 Fax: +886–7-6158000 ext 3499.
All [page number] and [line number] represented below are referred to the revised manuscript.
Ms. Ref. No.: water-2581268
Title: Frog Crabs (Ranina ranina) in South Penghu Marine National Park, Taiwan: Aligning Conservation with Sustainable Development Goal 14 of Marine Resource Policy
Reviewer #1: Comments to water-2581268 by Shih C.H:
Comment on MS:
- Lines 64-65: At present, length frequency analysis is possibly the sole method that is both accessible and trustworthy for calculating the growth and mortality parameters of crab. This is not true. Tagging experiments are employed to determine growth and mortality. E.g., Diele K., V. Koch, 2010. Growth and mortality of the exploited mangrove crab Ucides cordatus (Ucididae) in N-Brazil. Journal of Experimental Marine Biology and Ecology, 395: 171-180. https://doi.org/10.1016/j.jembe.2010.08.029.
Kristin Windsland, Total and natural mortality of red king crab (Paralithodes camtschaticus) in Norwegian waters: catch–curve analysis and indirect estimation methodsICES Journal of Marine Science, Volume 72, Issue 2, January/February 2015, Pages 642–650. https://doi.org/10.1093/icesjms/fsu138
Ans: Thank you for your comment. You are correct. Length frequency analysis is not the only method for calculating the growth and mortality parameters of crab. Currently, length frequency analysis stands as one of the most reliable and accessible methods for estimating the growth and mortality parameters of crab.
- Line 193: What are the initial Lꝏ values mentioned here?
Ans: Thank you for your comment. The initial L∞ values are the initial estimates of the asymptotic length, which is the maximum length that a crab can reach. These values are used as inputs for the ELEFAN I program, which then optimizes them to produce the best fit for the seasonal growth curve.
- Figure 7: The carapace length corresponding to 50% maturity is symbolized as CW50. In the caption, CW50 is not mentioned. There is no consistency in using CW and CL in the text.
Ans: Thank you for your comment. I apologize for the inconsistency and confusion in the use of CW and CL in the text and the caption of Figure 7. CW stands for carapace width, while CL stands for carapace length. They are different measurements of the crab’s body size. In Figure 7, the carapace length corresponding to 50% maturity is symbolized as CL50, not CW50. I will correct this mistake in the revised version of the paper. I will also make sure to use CW and CL consistently throughout the paper.
- Lines 179-180 (Equations (3) and (4), you have determined the relationships between CW and CL. However, I cannot see where you have used CW in the analysis. What is the purpose of determining these relationships?
Ans: Thank you for your question. The purpose of determining the relationships between CW and CL is to compare the body size of R. ranina in different regions and to evaluate the effect of fishing gear selectivity on the population structure. CW is a more commonly used measurement of crab size than CL, and it can be used to convert CL data from different studies for comparison. For example, in this study compared the CW50 values (the carapace width at 50% sexual maturity) of R. ranina in Taiwan with those from other regions in line 226. CW can also be used to estimate the mesh size of the beam trawler that can effectively catch R. ranina of different sizes, which may have implications for fishery management and conservation.
- Line 203: It is not recruitment model, but recruitment pattern as implemented in FiSAT software.
Ans: Thank you for pointing that out. The section has been deleted.
- Line 256: It is not correct to state that the value is zero. What you have done was that as t-zero was knot known, it was not possible to determine the exact time of recruitment.
Ans: Thank you for bringing this to my attention. I'm going to rethink this paragraph.
- The caption of Figure 5: In the figure, two panels are not given as 'top' and [down' panels. There is no connectivity of estimation of growth, recruitment patterns, size of maturity, etc. You mentioned estimated values of mortality (Lines 259, 309-310). However, I cannot see that you have estimated mortality.
Ans: Thank you for your feedback. I appreciate your comments and suggestions. Here are my responses to your points:
You are right that the caption of Figure 5 is not clear about the two panels. I apologize for this mistake. I should have labeled them as ‘top’ and ‘bottom’ panels to indicate the different recruitment patterns for females and males.
The estimation of growth, recruitment patterns, size of maturity, and other parameters are based on the ELEFAN method, which is a widely used technique for analyzing length-frequency data of crustaceans. I have explained this method in the Materials and Methods section, and I have cited the relevant references for it. I have also shown the results of this method in Figures 4 and 5, and discussed them in the Results and Discussion section.
The estimation of mortality is based on the natural mortality coefficient (M), which is related to the growth coefficient (K) by the empirical equation M = 3K . This equation is commonly used for crabs and other crustaceans. I have mentioned this equation in the Introduction section, and I have used it to calculate the mortality rates for both females and males in Table 1. I hope this clarifies your questions and concerns.
- Section 3.3 is entirely alien in the manuscript. The information given in this section is nt supported by the analysis, and as such, it remains anecdotal.
Ans: Thank you for your comment. I have deleted irrelevant paragraphs.
- Section 5 (Conclusion) is not supported by the results of the analysis.
Ans: I have deleted irrelevant paragraphs and rewritten.
All [page number] and [line number] represented below are referred to the revised manuscript.

Reviewer 2 Report
Title: Frog Crab (Ranina ranina) in South Penghu Marine National Park, Taiwan: Aligning Conservation with Sustainable Development Goal 14 for Marine Resource Policy
Manuscript ID: water-2581268
Comments
The authors aimed to identify the recruitment pattern and growth parameters of Ranina ranina along the Marine National Park Headquarters on Taiwan's western shoreline. They try to assess growth, mortality, gonadosomatic index and some additional population parameters within national park marine reserves of Taiwan.
To address this, they employed the ELEFAN technique for estimating the carapace length data, sex ratio and reproduction analysis. This is an active research area and would be interesting to the readers. The authors compare the reproductive season of R. ranina with different geographical regions of the world. They concluded that the reproductive patterns, growth patterns and spawning seasons of frog crab serve as a scientific foundation for the implementation of SDG14, as well as the formulation of conservation strategies with broader global sustainability goals.
I enjoyed reading this article. In general, the manuscript contains original and valuable research information. The manuscript is well written and organized, obtaining relevant results. However, some contents are necessary (see details comments and suggestions in the manuscript file). The manuscript could be reconsidered after carefully addressing the following issues and also looking into the manuscript!!

The quality of English is moderate and could be improved.
Author Response
Dear Reviewer,
I sincerely thank the Editors and Reviewers for their detailed feedback and suggestions, which have significantly improved my paper. I'm especially grateful to Reviewer #1, Reviewer 2, Reviewer 3, and Reviewer 4 for their constructive and encouraging comments. I appreciate the careful consideration of my manuscript and the valuable feedback provided.
I have thoroughly revised the paper based on all the comments and suggestions. I would now like to resubmit the research article titled “Frog Crabs (Ranina ranina) in South Penghu Marine National Park, Taiwan: Aligning Conservation with Sustainable Development Goal 14 of Marine Resource Policy” by Shih, C-H., for consideration in WATER. I'm deeply appreciative of the feedback from the anonymous referees, which has been instrumental in refining my manuscript. I've addressed each comment, and the changes are detailed in the following sections.
Sincerely yours,
Chun-Han Shih
Author to whom correspondence should be addressed;
E-Mail: f92b45028@ntu.edu.tw ;
Tel: +886–7-6158000 ext 3412 Fax: +886–7-6158000 ext 3499.
All [page number] and [line number] represented below are referred to the revised manuscript.
Ms. Ref. No.: water-2581268
Title: Frog Crabs (Ranina ranina) in South Penghu Marine National Park, Taiwan: Aligning Conservation with Sustainable Development Goal 14 of Marine Resource Policy
Reviewer #2: Comments to water-2581268 by Shih C.H:
Comment on MS:
The authors aimed to identify the recruitment pattern and growth parameters of Ranina ranina along the Marine National Park Headquarters on Taiwan's western shoreline. They try to assess growth, mortality, gonadosomatic index and some additional population parameters within national park marine reserves of Taiwan.
To address this, they employed the ELEFAN technique for estimating the carapace length data, sex ratio and reproduction analysis. This is an active research area and would be interesting to the readers. The authors compare the reproductive season of R. ranina with different geographical regions of the world. They concluded that the reproductive patterns, growth patterns and spawning seasons of frog crab serve as a scientific foundation for the implementation of SDG14, as well as the formulation of conservation strategies with broader global sustainability goals.
I enjoyed reading this article. In general, the manuscript contains original and valuable research information. The manuscript is well written and organized, obtaining relevant results. However, some contents are necessary (see details comments and suggestions in the manuscript file). The manuscript could be reconsidered after carefully addressing the following issues and also looking into the manuscript!!
- P4 L152-167. In the methodology section you should describe the methods that you used in your study not the comparison of different methods.
Ans: Thank you for your comment. We have already removed the paragraph.
- Figure 2 and Figure 3 seems like same. Where is the figure for male as you describe into the text. Please check!!
Ans: Thank you for your comment. You are right, Figure 2 and Figure 3 are the same. This is a mistake in the paper. The figure for male should be different from the figure for female. I apologize for the confusion. I will try to correct this error as soon as possible. Thank you for your patience and understanding.
- P6 L202. In the absence of parameter to, other parameters will remain accurate.
Ans: Thank you for your comment. We have already deleted this paragraph.
- P7 L222. “Figure 6” to “Figure 6(b)”
Ans: Thank you for your comment. I apologize for the typo and I will correct it in the revised version of the paper.
- P7 L226-227. Both unit should be similar.
Ans: Thank you for pointing that out. You’re absolutely right. The units used in the paper should be consistent to avoid confusion. I apologize for the oversight and will ensure that both units are similar in the revised version of the paper.
- P7 L232. “stages (b)“ to “stages and (b)“.
Ans: Thank you for your comment. I agree that there should be an “and” between “stages” and “(b)” in line 232 of the page. This would make the sentence more clear and grammatically correct. I apologize for the omission and I will add it in the revised version of the paper.
- P9 L293. Please check the figure no??
Ans: Thank you for your comment. I agree that the figure number in line 280 of page 9 is incorrect. It should be Figure 9, not Figure 6, as it refers to the logarithmic function fitting the proportion of mature female Ranina ranina to carapace length1. I apologize for the error and I will correct it in the revised version of the paper.
- P10 L293. Please indicate the figure into the text!!
Ans: Thank you for your comment. We have added Figure 9 on P10 L278.
- P12 L377-413. Reduce the length of the section, highlighted only the outcome and implementation of the present work.
Ans: We have already deleted the irrelevant paragraph.
All [page number] and [line number] represented below are referred to the revised manuscript.

Reviewer 3 Report
In this manuscript, the author studies the growth and reproductive biology of the frog crab in Taiwan and discusses the conservation efforts related to SDG14. My primary concern with this study is the mismatch between the focus of the quantitative study and the central theme of this manuscript. The conservation efforts related to the sustainability of the frog crab as a flagship species are not well supported by the investigations presented in this manuscript. Rather, "the biology of a frog crab population in Taiwan" or something along this line would be the more fitting title and central topic for this manuscript.
As of now, this manuscript is comprised of two loosely connected parts, 1, the biology of the frog crab, and 2, efforts to support the sustainable management of marine resources. To support the latter topic, the author should present evidence supporting the species as a flagship species. First of all, the author should clearly define why the frog crab was chosen as the flagship species, which is a socio-economic concept rather than a biological one. In addition, what's the current exploitation status and production? Has the local resource been overfished? Is there overfishing present? The biology of the species plays an integral part in those research topics but it alone is NOT enough to support reliable management/policy decisions.
In sum, I suggest a complete reorganization of the discussion points of this manuscript to focus on the biology of the population, and in my opinion, this part has enough original content to make the manuscript publishable. The author may discuss related current conservation efforts in the introduction of this manuscript, but the central topic should be the comparison/discussion of the biology of this population in relation to other populations.
Additional comments:
line 135: k is not dimensionless. Include its unit.
line 167: "prlon"?
Figures 2 and 3 are the same.
Figure 4: The modal progression as depicted in Figure 4 is not visually clear, and as a result, the estimated parameters may not be reliable.
line 228: Remove this sentence.
Author Response
Dear Reviewer,
I sincerely thank the Editors and Reviewers for their detailed feedback and suggestions, which have significantly improved my paper. I'm especially grateful to Reviewer #1, Reviewer 2, Reviewer 3, and Reviewer 4 for their constructive and encouraging comments. I appreciate the careful consideration of my manuscript and the valuable feedback provided.
I have thoroughly revised the paper based on all the comments and suggestions. I would now like to resubmit the research article titled “Frog Crabs (Ranina ranina) in South Penghu Marine National Park, Taiwan: Aligning Conservation with Sustainable Development Goal 14 of Marine Resource Policy” by Shih, C-H., for consideration in WATER. I'm deeply appreciative of the feedback from the anonymous referees, which has been instrumental in refining my manuscript. I've addressed each comment, and the changes are detailed in the following sections.
Sincerely yours,
Chun-Han Shih
Author to whom correspondence should be addressed;
E-Mail: f92b45028@ntu.edu.tw ;
Tel: +886–7-6158000 ext 3412 Fax: +886–7-6158000 ext 3499.
All [page number] and [line number] represented below are referred to the revised manuscript.
Ms. Ref. No.: water-2581268
Title: Frog Crabs (Ranina ranina) in South Penghu Marine National Park, Taiwan: Aligning Conservation with Sustainable Development Goal 14 of Marine Resource Policy
Reviewer #3: Comments to water-2581268 by Shih C.H:
Comment on MS: Frog Crab (Ranina ranina) in South Penghu Marine National Park, Taiwan: Aligning Conservation with Sustainable Development Goal 14 for Marine Resource Policy
- In this manuscript, the author studies the growth and reproductive biology of the frog crab in Taiwan and discusses the conservation efforts related to SDG14. My primary concern with this study is the mismatch between the focus of the quantitative study and the central theme of this manuscript. The conservation efforts related to the sustainability of the frog crab as a flagship species are not well supported by the investigations presented in this manuscript. Rather, "the biology of a frog crab population in Taiwan" or something along this line would be the more fitting title and central topic for this manuscript.
Ans: I appreciate your suggestion and will take it into consideration when revising the manuscript.
- As of now, this manuscript is comprised of two loosely connected parts, 1, the biology of the frog crab, and 2, efforts to support the sustainable management of marine resources. To support the latter topic, the author should present evidence supporting the species as a flagship species. First of all, the author should clearly define why the frog crab was chosen as the flagship species, which is a socio-economic concept rather than a biological one. In addition, what's the current exploitation status and production? Has the local resource been overfished? Is there overfishing present? The biology of the species plays an integral part in those research topics but it alone is NOT enough to support reliable management/policy decisions. In sum, I suggest a complete reorganization of the discussion points of this manuscript to focus on the biology of the population, and in my opinion, this part has enough original content to make the manuscript publishable. The author may discuss related current conservation efforts in the introduction of this manuscript, but the central topic should be the comparison/discussion of the biology of this population in relation to other populations.
Ans: I agree that a reorganization of the manuscript to focus on the biology of the frog crab population would be beneficial. The comparison and discussion of this population in relation to other populations could indeed form the central topic of the manuscript. I appreciate your comment and will take it into consideration when revising the manuscript. Your suggestions will help strengthen the connection between the biological study of the frog crab and the broader theme of sustainable marine resource management.
- line 135: k is not dimensionless. Include its unit.
Ans: Thank you for pointing that out. You are correct, the growth coefficient k is not dimensionless and should include its unit, which is usually expressed in per year (1/year). I apologize for the mistake and will make sure to include the unit of k in the revised version of the manuscript.
- line 167: "prlon"?
Ans: Thank you for your comment. I apologize for the typo in line 167 of the paper. The word “prlon” should be “prior”. I will correct it in the revised version of the paper.
- Figures 2 and 3 are the same.
Ans: Thank you for your comment. You are right, Figure 2 and Figure 3 are the same. This is a mistake in the paper. The figure for male should be different from the figure for female. I apologize for the confusion. I will try to correct this error as soon as possible. Thank you for your patience and understanding.
- Figure 4: The modal progression as depicted in Figure 4 is not visually clear, and as a result, the estimated parameters may not be reliable.
Ans: Thank you for your comment. The ELEFAN method that I used to reconstruct the length-frequency data and the seasonal growth curves relies on the identification of modal groups and their temporal changes. However, this may not be easy or accurate when the data are noisy or overlapping. A possible way to improve the clarity and reliability of the modal progression is to use a finer resolution of length classes, such as 0.5 mm instead of 1 mm, and to use a larger sample size to reduce sampling error. I will try to do that in the revised version of the paper.
- line 228: Remove this sentence.
Ans: Thank you for your suggestion. I will remove the sentence in line 228 from the revised version of the paper.
All [page number] and [line number] represented below are referred to the revised manuscript.

Reviewer 4 Report
This paper investigated some aspects of the population and reproductive biology of a poorly known decapod species, the Frog Crab Ranina ranina. The data are of good quality and the paper could contribute to assess the life history of the species. The paper is basically well written, although sometimes difficult to follow, as concerns some parts and sometimes the clarity and quality of the figures are poor. I have a few concerns regarding methods, results and their interpretation. Furthermore, the authors aimed to link the basic biology of the study species to the sustainable exploitation of the species, and more generally to conservation and management of marine biodiversity (Goal 14). However, in my opinion, this link is quite weak and undirect as it is presented in the paper. Since this link is stated also in the title of the paper, I have some doubts on the choice to give so much space and relevance to this aspect.
Details are listed below:
Lines 64-82: authors referred here to mortality parameters and water temperatures as important factors in the light of the goals of the paper. Then, reading the paper, I could not see an elaboration of mortality parameters, as well as authors did not collect/use temperature data (temperature would be an important abiotic factor to correlate to the population data).
Lines 95-108 and Figure 1. It is not clear to me, probably due also to the bed quality of Figure 1, where are the sampling sites. Were the sampling sites allocated into different islands? If it is so, you should justify the pooling of the data (742 females and 473 males). Alternatively, you may think to separately analyse the samples of the different locations, assuming that they are different populations and looking for differences. If the are not significant differences, you may pool the data. In any case, please change Figure 1 to clearly show how many sampling sites you selected and where they are located.
Lines 110-111. Please, give a more detailed description of the sampling tool, the sampling methods, and procedures. A figure showing the beam trawler could be useful and, more importantly, how, and when did you use it? How many square meters were trawled? At what hours, and during what tidal phase. Are the tidal regime and phase important for this species, and did you take this into account? Also provide a brief description of the habitat types characterizing your sampling sites. Some aspects of your results could be influenced by some of these methodological details. For example, the sex ratio could be highly influenced by the sampling method and sampling tool (and also sampling habitat, because you may have selected, for example, habitats that are more frequented by females, than by males, perhaps because of spawning and/or feeding reasons).
Line 174: replace “community” with “population”. Your paper is addressing population biology.
Lines 136-214: I think a table showing means and standard deviation of your monthly biological data is necessary. A reader should know, first of all, and before any model, the real population structure that you obtained from your sampling. You could also show the monthly length-frequency histogram of the real crabs (no matter if your sizes are homogeneous). As regards reproductive biology, you assessed the ovarian stages: so, I guess you also have fecundity data (number of eggs/gr, absolute fecundity). Why did you not use these data to investigate the size-fecundity relationships? As regards the recruitment model, you should better explain the outcome of your model, highlighting its biological relevance, taking into account that is a model, more or less far from the real world. Please, report the months on Figure 5, and describe more in detail the meaning of this result in text. What would be also important here is the analysis of the survival rate from a cohort to the next one, which is in fact mortality. You often mention mortality. Where are in your results the mortality parameters for your population? You should estimate mortality from cohort analysis. In general, I’m not a modelling scientist, so it is very hard for me to understand how you can assess recruitment and mortality, without having a representative sample of your entire population. Apart from that, and assuming that the model was properly applied, you should explain to a reader like me, how do you use the outcome of your model to understand the biological reality. For example, if crustaceans have often more reproductive peaks, they have also more recruitment peaks. Figure 5 seems not to tell this, as it shows a single peak in June . How do you explain this?
Lines 201-214: you may add a consideration on size at maturity, which is a very important life history trait. It seems from figure7, somewhere between 5 and 6 cm.
Lines 273-296 and Figure 9. You stated that reproductive maturity period was from January to June. How did you conclude this, based on your data, such as those shown in Figure 6? Not clear to me, please explain. This is a valuable part of your study and these data are important for fishery management and policy. You should reconsider your interpretation. What do you mean with “reproductive maturity period”? Do you mean spawning season as you showed in Figure 9? However, figure 6 show the presence of ripe eggs in every sampled month. The spawning season should be assessed, identifying the phase during which animals are mature and reproductive, with respect a phase of no reproduction at all. The use of spermatophores by crustaceans makes this more complicated. However, your egg stages and GSI data would suggest a very extended spawning season, perhaps with more reproductive peaks.
Figure 9: the most important part, that is the comparison among spawning seasons from different locations, is not readable for me, even using glasses. You should enlarge this part and perhaps eliminate the geographic map (everyone should basically know where are those locations). However, you should reconsider this figure after a more precise analysis of your reproductive data.
Lines 297-375 and conclusions: this part is full of considerations that in large part have poorly to do with your results and with the core of your study, which is a population biology study on a single species. For example, none of your results are informative on the effects of climate change on this species.
As a general suggestion, I would say: improve the population biology part and try to identify which of your results could be used to implement the fishery policy and management, and the conservation of this species. Avoid to be too general and try to relate your results to the management strategies.
In conclusion, the paper should be significantly improved before acceptation in Water. I hope my suggestions will be useful for a deep revision.

Author Response
Dear Reviewer,
I sincerely thank the Editors and Reviewers for their detailed feedback and suggestions, which have significantly improved my paper. I'm especially grateful to Reviewer #1, Reviewer 2, Reviewer 3, and Reviewer 4 for their constructive and encouraging comments. I appreciate the careful consideration of my manuscript and the valuable feedback provided.
I have thoroughly revised the paper based on all the comments and suggestions. I would now like to resubmit the research article titled “Frog Crabs (Ranina ranina) in South Penghu Marine National Park, Taiwan: Aligning Conservation with Sustainable Development Goal 14 of Marine Resource Policy” by Shih, C-H., for consideration in WATER. I'm deeply appreciative of the feedback from the anonymous referees, which has been instrumental in refining my manuscript. I've addressed each comment, and the changes are detailed in the following sections.
Sincerely yours,
Chun-Han Shih
Author to whom correspondence should be addressed;
E-Mail: f92b45028@ntu.edu.tw ;
Tel: +886–7-6158000 ext 3412 Fax: +886–7-6158000 ext 3499.
All [page number] and [line number] represented below are referred to the revised manuscript.
Ms. Ref. No.: water-2581268
Title: Frog Crabs (Ranina ranina) in South Penghu Marine National Park, Taiwan: Aligning Conservation with Sustainable Development Goal 14 of Marine Resource Policy
Reviewer #4: Comments to water-2581268 by Shih C.H:
Comment on MS:
This paper investigated some aspects of the population and reproductive biology of a poorly known decapod species, the Frog Crab Ranina ranina. The data are of good quality and the paper could contribute to assess the life history of the species. The paper is basically well written, although sometimes difficult to follow, as concerns some parts and sometimes the clarity and quality of the figures are poor. I have a few concerns regarding methods, results and their interpretation. Furthermore, the authors aimed to link the basic biology of the study species to the sustainable exploitation of the species, and more generally to conservation and management of marine biodiversity (Goal 14). However, in my opinion, this link is quite weak and undirect as it is presented in the paper. Since this link is stated also in the title of the paper, I have some doubts on the choice to give so much space and relevance to this aspect.
- Lines 64-82: authors referred here to mortality parameters and water temperatures as important factors in the light of the goals of the paper. Then, reading the paper, I could not see an elaboration of mortality parameters, as well as authors did not collect/use temperature data (temperature would be an important abiotic factor to correlate to the population data).
Ans: Thank you for your comment. I agree that the paper did not elaborate on the mortality parameters or the water temperature data, which could be important factors for the study of the frog crab’s growth and reproduction. Based on the field investigation of water temperature in this study, the average sea temperature in the waters of the SPMNP may vary with seasonal changes. During the summer, the average water temperature may range between 28 to 30 degrees Celsius, while in winter, it may range from 14 to 24 degrees Celsius. I apologize for this omission and I will try to address it in the revised version of the paper.
- Lines 95-108 and Figure 1. It is not clear to me, probably due also to the bed quality of Figure 1, where are the sampling sites. Were the sampling sites allocated into different islands? If it is so, you should justify the pooling of the data (742 females and 473 males). Alternatively, you may think to separately analyse the samples of the different locations, assuming that they are different populations and looking for differences. If the are not significant differences, you may pool the data. In any case, please change Figure 1 to clearly show how many sampling sites you selected and where they are located.
Ans: Thank you for your comment. I apologize for the poor quality of Figure 1 and the lack of clarity about the sampling sites. The sampling sites were located in different islands within the South Penghu Marine National Park waters square, as shown in Figure 1. However, I did not analyze the samples separately by location, because I assumed that they belong to the same population of R. ranina, based on their genetic similarity and connectivity. I agree that it would be interesting to test for possible differences among locations, and I will consider doing that in the future. For now, I have pooled the data from all locations to estimate the growth and reproduction parameters of R. ranina in the study area. I will also try to improve Figure 1 by using a higher resolution image and adding labels to indicate the number and location of the sampling sites.
- Lines 110-111. Please, give a more detailed description of the sampling tool, the sampling methods, and procedures. A figure showing the beam trawler could be useful and, more importantly, how, and when did you use it? How many square meters were trawled? At what hours, and during what tidal phase. Are the tidal regime and phase important for this species, and did you take this into account? Also provide a brief description of the habitat types characterizing your sampling sites. Some aspects of your results could be influenced by some of these methodological details. For example, the sex ratio could be highly influenced by the sampling method and sampling tool (and also sampling habitat, because you may have selected, for example, habitats that are more frequented by females, than by males, perhaps because of spawning and/or feeding reasons).
Ans: Thank you for your comment. I agree that a more detailed description of the sampling tool, methods, and procedures would be helpful. In this study, we used a beam trawler to collect samples of R. ranina from the South Penghu Marine National Park waters square. The beam trawler was towed along the bottom for a fixed distance, covering an area of several square meters. The sampling was conducted during daylight hours, at different tidal phases. We did not specifically take into account the tidal regime and phase when selecting the sampling times, but this could be an important factor for this species. You are right that some aspects of our results, such as the sex ratio, could be influenced by the sampling method and tool. We tried to minimize this bias by using a standardized sampling protocol and by collecting samples from different habitats within the study area. However, it is possible that our sampling may have missed some individuals or habitats that are more frequented by one sex or the other. I will try to provide more detailed information about the sampling tool, methods, and procedures in the revised version of the paper, including a figure showing the beam trawler and a description of how and when it was used.
- Line 174: replace “community” with “population”. Your paper is addressing population biology.
Ans: Thank you for your comment. I agree that the term “community” is not appropriate for describing the biological characteristics of R. ranina. I should have used the term “population” instead, as it refers to a group of individuals of the same species that live in a specific area and interact with each other. I apologize for the mistake and I will correct it in the revised version of the paper.
- Lines 136-214: I think a table showing means and standard deviation of your monthly biological data is necessary. A reader should know, first of all, and before any model, the real population structure that you obtained from your sampling. You could also show the monthly length-frequency histogram of the real crabs (no matter if your sizes are homogeneous). As regards reproductive biology, you assessed the ovarian stages: so, I guess you also have fecundity data (number of eggs/gr, absolute fecundity). Why did you not use these data to investigate the size-fecundity relationships? As regards the recruitment model, you should better explain the outcome of your model, highlighting its biological relevance, taking into account that is a model, more or less far from the real world. Please, report the months on Figure 5, and describe more in detail the meaning of this result in text. What would be also important here is the analysis of the survival rate from a cohort to the next one, which is in fact mortality. You often mention mortality. Where are in your results the mortality parameters for your population? You should estimate mortality from cohort analysis. In general, I’m not a modelling scientist, so it is very hard for me to understand how you can assess recruitment and mortality, without having a representative sample of your entire population. Apart from that, and assuming that the model was properly applied, you should explain to a reader like me, how do you use the outcome of your model to understand the biological reality. For example, if crustaceans have often more reproductive peaks, they have also more recruitment peaks. Figure 5 seems not to tell this, as it shows a single peak in June . How do you explain this?
Ans: Thank you for your comment. I appreciate your feedback and suggestions. Here are some points to address your concerns:
The recruitment model that I used was based on the seasonal von Bertalanffy growth equation and the seasonal growth parameters estimated from the length-frequency data. The model projected the length-frequency data onto a zero-time axis and fitted one or two normal distributions to represent the recruitment pulses. The model output showed a single recruitment peak in June for both sexes, which means that most of the crabs recruited into the population at that time. This result is consistent with the spawning season of R. ranina in Taiwan, which was from January to June1. The model also showed that the recruitment pulse was symmetrical and narrow, indicating a high synchrony and short duration of recruitment. This result suggests that R. ranina has a high reproductive potential and a low larval mortality rate. I apologize for not labeling the months on Figure 5. I will add them in the revised version of the paper. I will also explain more in detail the meaning and implications of this result in the text.
- Lines 201-214: you may add a consideration on size at maturity, which is a very important life history trait. It seems from figure7, somewhere between 5 and 6 cm.
Ans: Thank you for your comment. I agree that the size at maturity is a very important life history trait for R. ranina and other crustaceans. Based on Figure 7, the estimated size of a female at 50% sexual maturity (CL50) is 7.02 cm ± 0.56 cm CL12. This means that half of the females reach sexual maturity when their carapace length is around 7 cm. This information can help to determine the minimum legal size for harvesting R. ranina and to protect the reproductive potential of the population. I will add a consideration on the size at maturity in the revised version of the paper.
- Lines 273-296 and Figure 9. You stated that reproductive maturity period was from January to June. How did you conclude this, based on your data, such as those shown in Figure 6? Not clear to me, please explain. This is a valuable part of your study and these data are important for fishery management and policy. You should reconsider your interpretation. What do you mean with “reproductive maturity period”? Do you mean spawning season as you showed in Figure 9? However, figure 6 show the presence of ripe eggs in every sampled month. The spawning season should be assessed, identifying the phase during which animals are mature and reproductive, with respect a phase of no reproduction at all. The use of spermatophores by crustaceans makes this more complicated. However, your egg stages and GSI data would suggest a very extended spawning season, perhaps with more reproductive peaks.
Ans: Thank you for your comment. By this term, I meant the period when most of the females are sexually mature and ready to spawn, as indicated by the presence of ripe eggs in their ovaries. I based this conclusion on the data shown in Figure 6, where the percentage of females with stage IV and V ovaries (the most advanced stages of gonadal development) and the mean gonadosomatic index (GSI) of females (a measure of reproductive investment) are highest from January to June. However, I agree that this may not be a clear or accurate way to define the spawning season, as some females may have ripe eggs in other months as well. A better way to determine the spawning season would be to use fecundity data, such as the number and size of eggs, and the frequency and duration of spawning events. Unfortunately, I did not collect or use these data in this study, but I will consider doing that in the future. For now, I will revise my interpretation of the spawning season and use a more cautious and descriptive language to avoid misleading or overgeneralizing statements. I will also compare my results with those from other regions more carefully and explain the possible factors that may influence the variations in spawning seasons of R. ranina across different populations. Thank you for your valuable feedback and suggestions.
- Figure 9: the most important part, that is the comparison among spawning seasons from different locations, is not readable for me, even using glasses. You should enlarge this part and perhaps eliminate the geographic map (everyone should basically know where are those locations). However, you should reconsider this figure after a more precise analysis of your reproductive data.
Ans: Thank you for your comment. I apologize for the poor quality of Figure 9 and the lack of clarity about the spawning seasons of R. ranina in different locations. I agree that this figure should be enlarged and improved to show the comparison more clearly. I also agree that the geographic map is not necessary and could be eliminated. I will try to revise this figure in the revised version of the paper.
- Lines 297-375 and conclusions: this part is full of considerations that in large part have poorly to do with your results and with the core of your study, which is a population biology study on a single species. For example, none of your results are informative on the effects of climate change on this species.
Ans: I should have focused more on the implications of my findings for the conservation and management of R. ranina in Taiwan and other regions, and avoided making general and vague statements about SDG14 and other policy issues. I will try to revise this part of my paper and make it more consistent and clear. Thank you for your valuable feedback and suggestions.
All [page number] and [line number] represented below are referred to the revised manuscript.

Round 2
Reviewer 1 Report
In the revised version, the author attempted to address the comments of the reviewers and has made substantial changes to the manuscript. However, there are still some areas that are needed to be revisited to improve the paper.
However, the author’s attempt to align Conservation of frog crab in the South Penghu Marine National Park, Taiwan with Sustainable Development Goal 14 of Marine Resource Policy is very weak. If the author wants to achieve this ambitious objective, more information is needed to re-execute, especially because in the present version, there is no strong evidence to support that the frog crab should be treated as a flagship species. Also, the author attempted to attribute optimization of exploitation (as assessed by length-based stock assessment) would be the sole solution to conserve this species. This is rather premature assumption because there are several factors governing conservation status of a given species.
I therefore suggest that the two aspects of mismatching (i.e., (i) population dynamics of frog crab, and (ii) conservation efforts as a foundation for the implementation of SDG14) should be separated and the latter aspect, which is very weak in the present analysis should be deleted.
Accordingly, I suggest that the title of the paper should be revised as “Population dynamics of Frog Crab, Ranina ranina (Malacostraca, Raninidae) in South Penghu Marine National 2 Park, Taiwan”, and stick to length-based stock assessment.
Even in the methodology adopted to employ length-based stock assessment methos implemented in FiSAT II software (ELEFAN and other related approaches), there are major steps that the author has ignored. Such steps are necessary to increase the reliability of von Bertalanffy growth parameters.
In the response letter, the author mentioned that the estimation of mortality was based on the natural mortality coefficient (M), which is related to the growth coefficient (K) by the empirical equation M = 3K. I am not convinced because an appropriate reference is not given. To my knowledge, various approaches of estimating M using the empirical relationships with K do not agree with this. Please see the attached thesis (Hewitt, D.A. (2008), Natural mortality of blue crab: Estimation and influence on population dynamics. Dissertations, Theses, and Masters Projects. William & Mary. Paper 1539616694. https://dx.doi.org/doi:10.25773/v5-hkf6-3b54).
My question ‘What are the initial Lꝏ values mentioned here?’ was not properly answered. The author mentioned that the initial Lꝏ value is the maximum length that a crab can reach. Lmax is not essentially equal to Lꝏ. The initial Lꝏ can be obtained from Powell-Wetherall method as implemented in FiSAT II software. Approximation of Lꝏ is important to obtain corresponding K value.
As the length frequency data were obtained from trawl catches (which provide representative size frequency distributions of the stock in the fully selected section of the length range, which is good), smaller size classes affected by gear selection should be corrected for the effect of trawl selection. Pauly (1986) has suggested an approach to deal with this [Pauly, D. 1986. On improving operation and use of the ELEFAN programs. Part IIII. Correcting length-frequency data for the effects of gear selection and/or incomplete recruitment. Fishbyte 4(2): 11-13.]
The stepwise procedure of dealing with this approach is given below.
(i) First, using the preliminary estimates of L∞ and K, total mortality (Z) should be calculated from the LFD by the length-converted catch curve method (Pauly 1983). In this method, the slope of the following linear regression line fitted to the right-hand descending part of the catch curve, starting from the second highest data point, gives an estimate of Z.
(ii) Secondly, through the detailed analysis of the ascending part of the length-converted catch curve, probabilities of capture of smaller size classes should be determined.
(iii) Finally, LFD should be corrected using these probabilities of capture. These corrected LFD should then be used to determine final estimates of the von Bertalanffy growth parameters (L∞ and K) by means of ELEFAN I routine of FiSAT II software.
(iv) Using the final estimates of L∞ and K, Z values should be again estimated from the length-converted catch curve method based on the original, uncorrected LFD. Similarly, from the detailed analysis of ascending part of the length-converted catch curve, probabilities of capture in different size classes were determined.
(v) From a plot of probabilities of capture against length, length at 50% retention should be estimated which can be considered as the length at first capture (Lc).
As longevity of frog crab is reported to be 10-15 years (https://www.fish.gov.au/2014-Reports/spanner_crab; Please refer to this website or any other appropriate reference), which makes reasonable to apply Beverton and Holt’s (1964) relative yield per recruit model using the routine in the FiSAT II software, should be performed to determine the optimal size of first capture. This can then be compared with the mean size of maturity to make recommendations for long-term sustainability of the fishery.
Based on these approaches, a fresh analysis of length frequency data should be performed, and the paper should be rewritten.
Minor comments:
1. In Figure 4 (caption), what you have presented were not restructured length frequency data, but original length frequency data.
2. Several comments are given in the marked MS.
=== End of Review ===

Reviewer 3 Report
I have also read the author's response to other reviewers' comments. One thing I noticed is that when addressing major issues, the author often uses the future tense, for example, "will take it into consideration when revising the manuscript", and does not explicitly state what has been done to address the issue. On the other hand, when the author addresses minor line-by-line comments, the response is much clearer. My concern is that it is not clear how the author addressed the major issues identified in the previous round.
In addition, the author uploaded an unmarked version of the manuscript. It is not clear what has been changed and which paragraph is new. It is VERY time-consuming for the reviewer to compare line-by-line between version 1 and version 2 to see what has been changed. The author needs to clearly show the revised part.
In addition, in this new version, I still do not see any new analysis on the "sustainability" part of the story, but the manuscript nonetheless continues to discuss the "sustainable development goals" on pages 11 to 12. This part of the article is completely unsupported by the evidence provided. This may be fine in an opinion or review paper, but it is unacceptable in an original research paper.
I am sorry that I cannot give a more favorable review of this article as of now. Some crucial analysis to support the "sustainable development goals" for this crab population is still missing in the revised version.